# Hetero-type dual photoanodes for unbiased solar water splitting with extended light harvesting

Jin Hyun Kim[1,*], Ji-Wook Jang[2,3,*], Yim Hyun Jo[4], Fatwa F. Abdi[2], Young Hye Lee[1], Roel van de Krol[2] & Jae Sung Lee[3]

Metal oxide semiconductors are promising photoelectrode materials for solar water splitting due to their robustness in aqueous solutions and low cost. Yet, their solar-to-hydrogen conversion efficiencies are still not high enough for practical applications. Here we present a strategy to enhance the efficiency of metal oxides, hetero-type dual photoelectrodes, in which two photoanodes of different bandgaps are connected in parallel for extended light harvesting. Thus, a photoelectrochemical device made of modified $BiVO_4$ and $\alpha\text{-}Fe_2O_3$ as dual photoanodes utilizes visible light up to 610 nm for water splitting, and shows stable photocurrents of $7.0 \pm 0.2$ mA cm$^{-2}$ at 1.23 $V_{RHE}$ under 1 sun irradiation. A tandem cell composed with the dual photoanodes–silicon solar cell demonstrates unbiased water splitting efficiency of 7.7%. These results and concept represent a significant step forward *en route* to the goal of >10% efficiency required for practical solar hydrogen production.

[1] School of Environmental Science & Engineering, Department of Chemical Engineering, Pohang University of Science and Technology (POSTECH), Pohang 790-784, South Korea. [2] Helmholtz-Zentrum Berlin für Materialien und Energie GmbH, Institute for Solar Fuels, Hahn-Meitner-Platz 1, Berlin 14109, Germany. [3] School of Energy and Chemical Engineering, Ulsan National Institute of Science and Technology (UNIST), Ulsan 44919, South Korea. [4] Advanced Center for Energy, Korea Institute of Energy Research (KIER), Ulsan 44919, South Korea. * These authors contributed equally to this work. Correspondence and requests for materials should be addressed to R.v.d.K. (email: roel.vandekrol@helmholtz-berlin.de) or to J.S.L. (email: jlee1234@unist.ac.kr).

The first requirement for efficient conversion of photons to electrons using a semiconductor is its proper bandgap ($E_g$). Photons with energy smaller than $E_g$ pass through the semiconductor without being absorbed (non-absorption loss), while photons with energy in excess of $E_g$ lose a part of their energy by emitting phonons (that is, lattice vibrations and heat) on absorption (thermalization losses)[1]. These two fundamental losses could be minimized by a multi-junction approach as proven in photovoltaics. Thus, a large $E_g$ semiconductor absorbs first the high-energy photons of the solar spectrum, and a small $E_g$ semiconductor placed behind it utilizes the low-energy photons that transmit through the first absorber. As a result, the theoretical efficiency of a multi-junction solar cell can reach 68%, breaking the Shockley–Queisser limit of 31% for a single junction solar cell[1,2].

We attempt to apply this concept to solar water splitting by combining two well-established metal oxide photoanodes, $BiVO_4$ and $Fe_2O_3$. Virtues of these metal oxide semiconductors are good stability in aqueous solutions[3–6], low-cost and easy solution-based processing[7]. However, $BiVO_4$ could use only photons of $\lambda < 510$ nm due to its rather large $E_g$ (direct $\sim 2.6$ eV and indirect $\sim 2.4$ eV)[5,8–14]. In contrast, $Fe_2O_3$ has a nearly ideal $E_g$ of $\sim 2.0$ eV, and can utilize the longer-wavelength ($< 620$ nm) photons. Yet, its solar-to-hydrogen conversion efficiencies ($\eta_{STH}$) of photoelectrochemical (PEC) water splitting remains modest because of its notoriously poor electrical properties[15–18]. Hence, by combining these two semiconductors, different parts of the solar spectrum could be utilized. The common way to utilize different $E_g$'s for efficient light harvesting is to form a heterojunction[12,19,20]. Unfortunately, when $BiVO_4$ and $Fe_2O_3$ are combined to form a bilayer-type heterojunction, this composite photoanode shows an inferior PEC water oxidation performance relative to either of the single-component photoanodes as demonstrated in Supplementary Fig. 1. The ineffective $BiVO_4/Fe_2O_3$ heterojunction originates from their straddling (type I) band alignment instead of a desired staggering (type II) type, which prevents the holes generated in $Fe_2O_3$ from transferring to $BiVO_4$, and thus is not suitable for effective charge separation.

To enhance the light harvesting in two semiconductors of different $E_g$'s, here we introduce a concept of 'hetero-type dual photoelectrode (HDP)' containing two independent liquid–semiconductor junctions as presented in Fig. 1. By using the large $E_g$ semiconductor as the top absorber and the small $E_g$ semiconductor at the bottom, both thermalization and non-absorption losses could be minimized. The advantage of our HDP concept is that current-matching of the two electrodes is not required unlike multi-junction solar cells[9] and

heterojunction photoelectrodes. Thus, we can independently optimize each photoelectrode, and the performance of the HDP device becomes the simple sum of the individual performance of the two. As a result, the spectral range of light harvesting in the solar spectrum utilized for PEC water splitting is extended.

We note that the HDP concept is analogous to a natural photosynthesis by seaweeds (or marine algae). They develop varying colours depending on the depth of the sea that they inhabit due to the availability of different photons at different depths in the sea—red light can reach only shallow depths because of its low photon energy ($\lambda > 600$ nm) while blue light ($\lambda < 420$ nm) can penetrate deeper. Adapting to the availability of photons of varying wavelengths, the seaweed colony is capable of selective light utilization for photosynthesis by varying their habitat depth from green (Chlorophyta), yellow (Phaeophyceae) and then to red (Rhodophyta) as one goes deeper into the sea[21]. Instead of developing an 'ideal' single-light absorber that can handle all range of photons, the seaweeds develop smart 'wavelength-optimized' light absorbers for sequential light utilization. Such resemblance is rather interesting, as the other commonly used light utilization scheme—so called tandem cell also resembles PSII/PSI of natural photosynthesis in using multiple light absorbers.

Here we implement the concept of the HDP using $BiVO_4$ as the front and $\alpha$-$Fe_2O_3$ as the back photoanodes for PEC water splitting. To extract the best possible performance of the single photoanodes, we need bulk and surface modifications by doping and co-catalyst, respectively, to improve the charge separation/ transfer properties of these semiconductors. As a result, a PEC device made of modified $BiVO_4$ and $\alpha$-$Fe_2O_3$ as dual photoanodes shows stable photocurrents of $7.0 \pm 0.2$ mA cm$^{-2}$ at $1.23$ $V_{RHE}$ under 1 sun irradiation utilizing visible light up to 610 nm. A tandem cell composed of HDP–silicon solar cell demonstrates unbiased water splitting efficiency of 7.7%.

## Results

**Fabrication of photoanodes and their PEC properties.** The base $BiVO_4$ electrode was fabricated by a metal-organic deposition method. Then, 1 at% (optimized) Mo doping and partial reduction treatment were performed to increase mainly the bulk charge separation efficiency as reported in our recent work[8]. The reduction treatment via a borohydride decomposition method induced $V_o$.. defects that effectively increase the charge carrier density in $BiVO_4$ lattice. Hence, the partial reduction and extrinsic Mo doping treatments synergistically improve the $n$-type conductivity of $BiVO_4$. The other photoanode, haematite $Fe_2O_3$, was synthesized by a nitrate decomposition method. This photoanode was doped with 0.5 at% (optimized) Ti, and also partially reduced by the borohydride decomposition. As shown by scanning electron microscopy (SEM) images in Supplementary Figs 2 and 3, $BiVO_4$ and $Fe_2O_3$ have nanoporous, well-connected and non-isotropic morphologies with typical feature sizes of $\sim 50$ and $\sim 10$ nm, respectively. Finally, co-catalysts of NiOOH/ FeOOH and $Ni_2FeO_x$ with a thin $TiO_2$ overlayer were deposited on the surface of $BiVO_4$ and haematite, respectively, to improve the charge carrier injection efficiency to the electrolyte. The SEM images in Supplementary Figs 2 and 3 show that the co-catalyst nanoparticles are well dispersed on each semiconductor surface. The X-ray diffraction crystal structures of the photoanode films are standard monoclinic $BiVO_4$ and haematite $\alpha$-$Fe_2O_3$ as shown in Supplementary Fig. 4. The X-ray photoemission spectroscopy (XPS) spectra in Supplementary Fig. 5 show binding energies at typically expected positions for $BiVO_4$ and $Fe_2O_3$. See the Methods section and Supplementary Discussion for detailed procedures and discussion for synthesis, optimization and characterization of the two modified photoanodes.

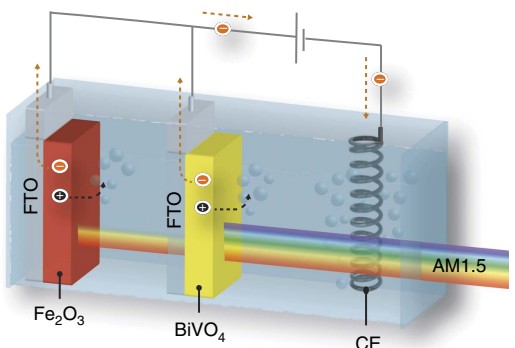

**Figure 1 | Wavelength-selective solar light absorption by hetero-type dual photoanode.** HDP made with different bandgap materials (for example, $BiVO_4$ and $Fe_2O_3$).

The bulk-modified films were tested first as photoanodes for PEC oxidation of a sacrificial sulphite ($SO_3^{2-}$) solution in a standard three-electrode PEC cell to evaluate their light absorption capability and bulk charge separation efficiency ($\eta_{bulk}$). As shown in Supplementary Fig. 6, doping with Mo (or Ti) combined with reduction treatment greatly increases the photocurrent and $\eta_{bulk}$. For $BiVO_4$, reduced 1% Mo:$BiVO_4$ shows $\eta_{bulk} \sim 90\%$ at 1.23 $V_{RHE}$, which is comparable to the reported state-of-the-art $BiVO_4$ photoanodes[6,8,22]. Doping and partial reduction treatments are also found to reduce the photocurrent gap between front and back illumination, indicating improved bulk electron transport[8,23]. The same tendency was also observed for $Fe_2O_3$, although the highest $\eta_{bulk}$ was only 41% at 1.23 $V_{RHE}$. The incident photon to current efficiency (IPCE) values of bare haematite above 400 nm in Supplementary Fig. 7 are relatively low as they originate from an inefficient indirect $d$–$d$ transition of $Fe^{3+}$ (ref. 18). But the Ti doping and reduction treatments are especially effective in increasing the IPCE of the indirect transition near the absorption edge. As demonstrated by the Mott–Schottky (MS) plots in Supplementary Fig. 8, the main effect of the doping and reduction treatments is to increase the charge carrier density of $BiVO_4$ and $Fe_2O_3$, and thereby to improve their bulk charge transfer properties.

Using these two optimized photoanodes, a HDP denoted as $BiVO_4\|Fe_2O_3$ was fabricated by placing the haematite electrode (H, $TiO_2$/0.5% Ti:$Fe_2O_3$) behind the $BiVO_4$ electrode (H, 1% Mo:$BiVO_4$) in parallel connection as depicted in Figs 1 and 2a. As shown in Fig. 2b, the fabricated $BiVO_4$, $Fe_2O_3$ and $BiVO_4\|Fe_2O_3$ electrodes are all highly transparent with a transmittance of 75% at 550 nm for $BiVO_4$, 70% at 650 nm for haematite and 50% at 650 nm for the HDP (Fig. 2f and Supplementary Fig. 9). Hence, $\sim 50\%$ of the incident light remains available for the third absorber (double c-Si) placed behind the HDP, which is needed to generate sufficient bias photovoltage to split water spontaneously as described below.

The PEC performance of the photoanodes (without co-catalysts) was investigated first in a three-electrode configuration (with Pt counter electrode) under AM1.5 illumination. An aqueous solution containing a sacrificial agent (0.5 KPi + 0.5 M $Na_2SO_3$) was used as an electrolyte. Since the photo-oxidation of sulphite is so facile, the hole injection efficiency ($\eta_{surface}$) is essentially 100%. Hence, we can compare bulk photoactivities of the photoanodes by these measurements while avoiding any complications of the hole injection at the semiconductor–electrolyte interfaces. As shown in Fig. 2c, photocurrents of 5.0 ± 0.2, 4.5 ± 0.2, 2.2 ± 0.1 and 7.1 ± 0.2 mA cm$^{-2}$ at 1.23 $V_{RHE}$ were obtained for $BiVO_4$, $Fe_2O_3$, $Fe_2O_3$ behind $BiVO_4$ (with no $BiVO_4$ contribution), and $BiVO_4\|Fe_2O_3$, respectively. These photocurrents are optimized values by adjusting the thickness of $BiVO_4$ and $Fe_2O_3$, which can be easily regulated by changing the quantity of precursor solution. (See Fig. 2d, Supplementary Figs 10 and 11 and Methods part for details).

Overall, the HDP shows an increased photocurrent by $\sim 29.3\%$ from that of a single $BiVO_4$ photoanode. Its photocurrent is almost the same as the sum of photocurrents generated by $BiVO_4$ and $Fe_2O_3$ behind $BiVO_4$, indicating that the HDP concept is indeed operating. It is also noteworthy that the onset potential of the $BiVO_4\|Fe_2O_3$ combination is $\sim 0.2$ $V_{RHE}$, which is the same as that of $BiVO_4$ alone and far lower than that of a single haematite photoanode ($\sim 0.4$ $V_{RHE}$).

The IPCE of $BiVO_4$ in Fig. 2e is significantly higher than that of haematite in the range of 300–450 nm, justifying $BiVO_4$ as the first absorber. The IPCE of the HDP further increases up to $\sim 95\%$ in this region due to the additional photocurrent generated from haematite using the transmitted light through the $BiVO_4$ electrode. In the range of 450–510 nm, IPCE of $BiVO_4$

drops sharply due to the reduced light-harvesting efficiency close to its absorption threshold and indirect band transition (direct: $\sim 2.6$ eV; indirect: $\sim 2.4$ eV)[14]. This is an important reason why the theoretical maximum photocurrent (7.5 mA cm$^{-2}$) of $BiVO_4$ is very difficult to achieve with a single $BiVO_4$ absorber. However, haematite can utilize the transmitted light in this region efficiently as an additional absorber. In the region of 510–610 nm, only haematite contributes to the overall performance of HDP. Even though the IPCE values in this region drop below 20%, the effect is very significant since more than 50% of the solar energy exploitable by a photoelectrode lies in this region. The theoretical maximum photocurrent of $Fe_2O_3$ is $\sim 13.6$ mA$^{-2}$, and nearly a half of it comes from photons with 500–620 nm range[24]. Thus, this region has to be exploited to further improve the performance of HDP.

**PEC water splitting with an external bias**. The performance of the photoanodes is now studied for water oxidation under 1 sun illumination in a three-electrode PEC cell without any sacrificial agent. When used as a single photoanode, electrolytes of pH 7 and >12 are optimal for $BiVO_4$ and $Fe_2O_3$, respectively, for their high activity and stability. Hence, the first important task to apply the HDP for PEC water splitting is to find an optimum electrolyte suitable for both $BiVO_4$ and $Fe_2O_3$. In a screening test summarized in Supplementary Figs 12 and 14, we found that pH 9.2 (1.0 M bicarbonate KCi solution) was a good compromise for the two semiconductors and selected co-catalysts. Next step is to find the best co-catalysts to promote the PEC water oxidation activity and stability by accelerating the hole injection to the electrolyte to avoid surface charge recombination, and to protect the semiconductor from photochemical corrosion. Thus, the co-catalyst selection process shown in Supplementary Figs 12–14 and described in the Supplementary Information yielded NiOOH/FeOOH for $BiVO_4$ and $Ni_2FeO_x$/$TiO_2$ for $Fe_2O_3$ as the best oxygen evolution co-catalysts. In particular, the thin $TiO_2$ layer deposited on the surface of $Fe_2O_3$ passivates the surface states of haematite (Supplementary Fig. 13) that cause undesired electron–hole recombination and potential drop within the Helmholtz layer[25]. Hence, the treatment not only improves photocurrent generation but also brings a negative shift of the onset potential and a greater photovoltage from the electrode.

The water oxidation photocurrents of completed $BiVO_4$, $Fe_2O_3$ and $Fe_2O_3$ behind $BiVO_4$ photoanodes were 5.0 ± 0.1, 4.0 ± 0.2 and 2.0 ± 0.1 mA cm$^{-2}$ at 1.23 $V_{RHE}$, respectively. The $BiVO_4\|Fe_2O_3$ HDP photoanode recorded stable photocurrents of 7.0 ± 0.2 and 8.0 ± 0.2 mA cm$^{-2}$ at 1.23 $V_{RHE}$ and 1.5 $V_{RHE}$, respectively. This PEC performance represents the highest ever reported for stable metal oxide photoelectrodes to the best of our knowledge. The benchmarking results against other studies are presented in Supplementary Table 1, in which it can be seen that not only HDP but also the individual $BiVO_4$ and $Fe_2O_3$ photoanodes are among the top performers within their classes. The performance was highly reproducible as demonstrated in Supplementary Fig. 15 and most of the photocurrents came from water splitting to $2H_2 + O_2$ gases with almost complete faradaic efficiency (Fig. 3d). The HDP utilized visible light up to 610 nm for water splitting as shown by action spectra in Fig. 3e–g. The photocurrents and IPCEs for water oxidation show similar values to those in Fig. 2 for sacrificial sulphite oxidation, indicating that used co-catalysts are highly effective, with hole injection efficiencies ($\eta_{surface}$) of 89–100%.

**Unbiased solar water-splitting tandem cell**. Eventually, the developed HDP must be successfully applied to a practical solar water splitting system that operates without external energy supply. Thus, unbiased solar water splitting was demonstrated

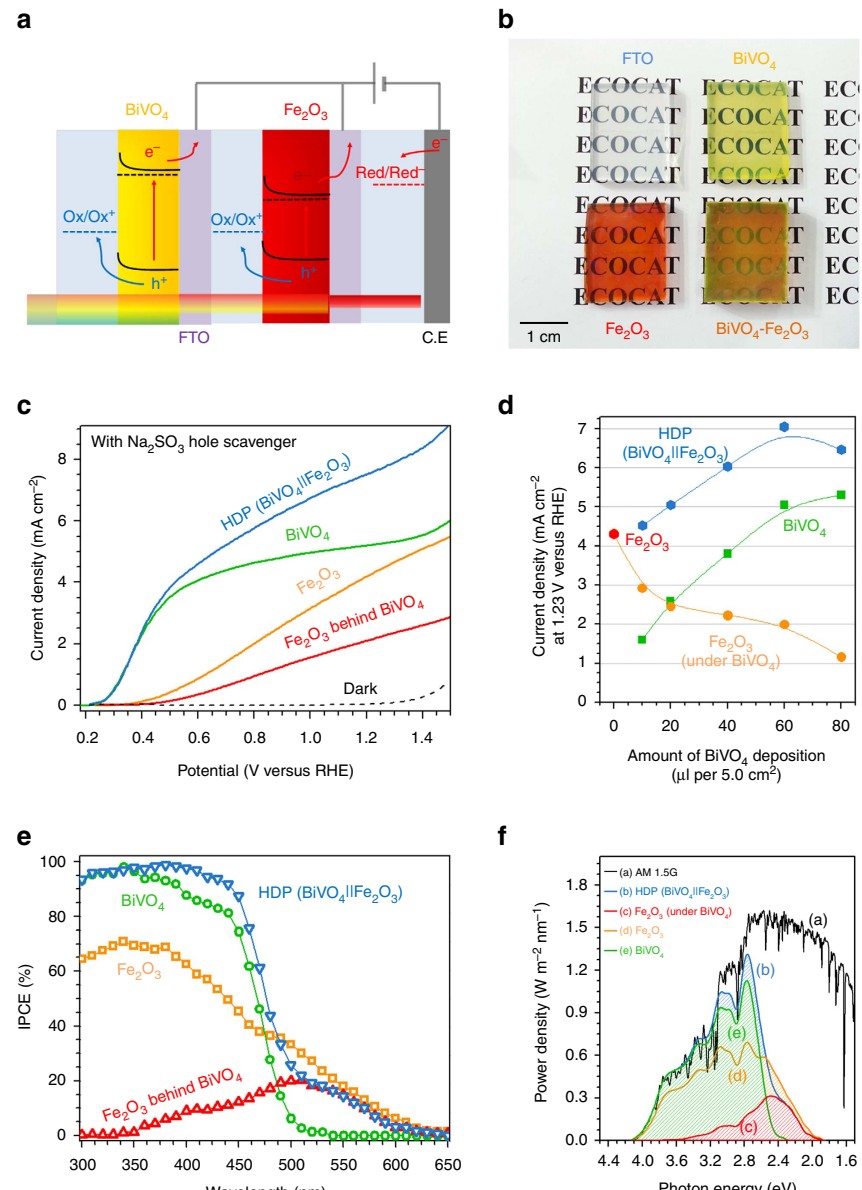

**Figure 2 | BiVO$_4$ and Fe$_2$O$_3$ HDP (BiVO$_4$||Fe$_2$O$_3$) in a sacrificial sulphite solution. (a)** Schematic working principle of a PEC cell with HDP BiVO$_4$||Fe$_2$O$_3$ as the photoanode. **(b)** Photographs of fabricated films. **(c)** Steady-state I–V curves of BiVO$_4$, Fe$_2$O$_3$, Fe$_2$O$_3$ behind BiVO$_4$, and BiVO$_4$||Fe$_2$O$_3$. **(d)** Optimization of the loading amount of BiVO$_4$ and Fe$_2$O$_3$ precursor solution (μl) on 5.0 cm$^2$ of FTO glass for the highest photocurrent generation (Supplementary Fig. 11). **(e)** IPCE. **(f)** Utilization of light in AM 1.5G spectrum by different photoanodes. Analyses were conducted in 0.5 M KPi and 0.5 M Na$_2$SO$_3$ of pH = 7.0.

with a tandem cell fabricated by integrating a thin film Si solar cell behind the HDP BiVO$_4$||Fe$_2$O$_3$ photoanode as schematically shown in Fig. 4a. We also fabricated monolithic version of the tandem cell (artificial leaf) that could perform overall water splitting (Fig. 4b, Supplementary Fig. 16 and Supplementary Movies 1 and 2 operating under simulated and actual outdoor sun illumination). The monolithic device possesses simplicity and versatility for small scale or mobile applications[24]. By using a series-connected two parallel (2p) c-Si solar cell with a photovoltage of ∼1.2 V as the bottom absorber, this system can be classified as a 'Q6' system, that is, quadruple absorbers that need six photons for production of a H$_2$ molecule[8,11,17,24,26]. The crossing points of performance curves of photoanodes and the solar cell give the expected operation photocurrents of BiVO$_4$, Fe$_2$O$_3$ and BiVO$_4$||Fe$_2$O$_3$ photoanodes at 4.5, 3.2 and 6.3 mA cm$^{-2}$, which correspond to solar-to-hydrogen conversion

efficiency ($\eta_{STH}$) of 5.6%, 3.9% and 7.7% (Fig. 4c, Supplementary Figs 17 and 18), respectively. As shown in Supplementary Fig. 18, the actual tandem cell reproduces these expected values, which represent one of the highest $\eta_{STH}$ obtained from unbiased solar water splitting in a tandem PEC cell using stable metal oxide photoelectrodes (see Supplementary Table 2 and Supplementary Fig. 19 for benchmarking)[8,9,27,28]. There was no sign of decay during the continuous run (8 h) of the tandem cell, indicating a complete isolation of the solar cell component from the electrolyte (Fig. 4d). Thus, we present the highest performing HDP photoanode and tandem cell for unbiased solar water-splitting system.

**Discussion**

In summary, we presented a HDP as a new strategy to fabricate a stand-alone, highly efficient solar water-splitting system. With a

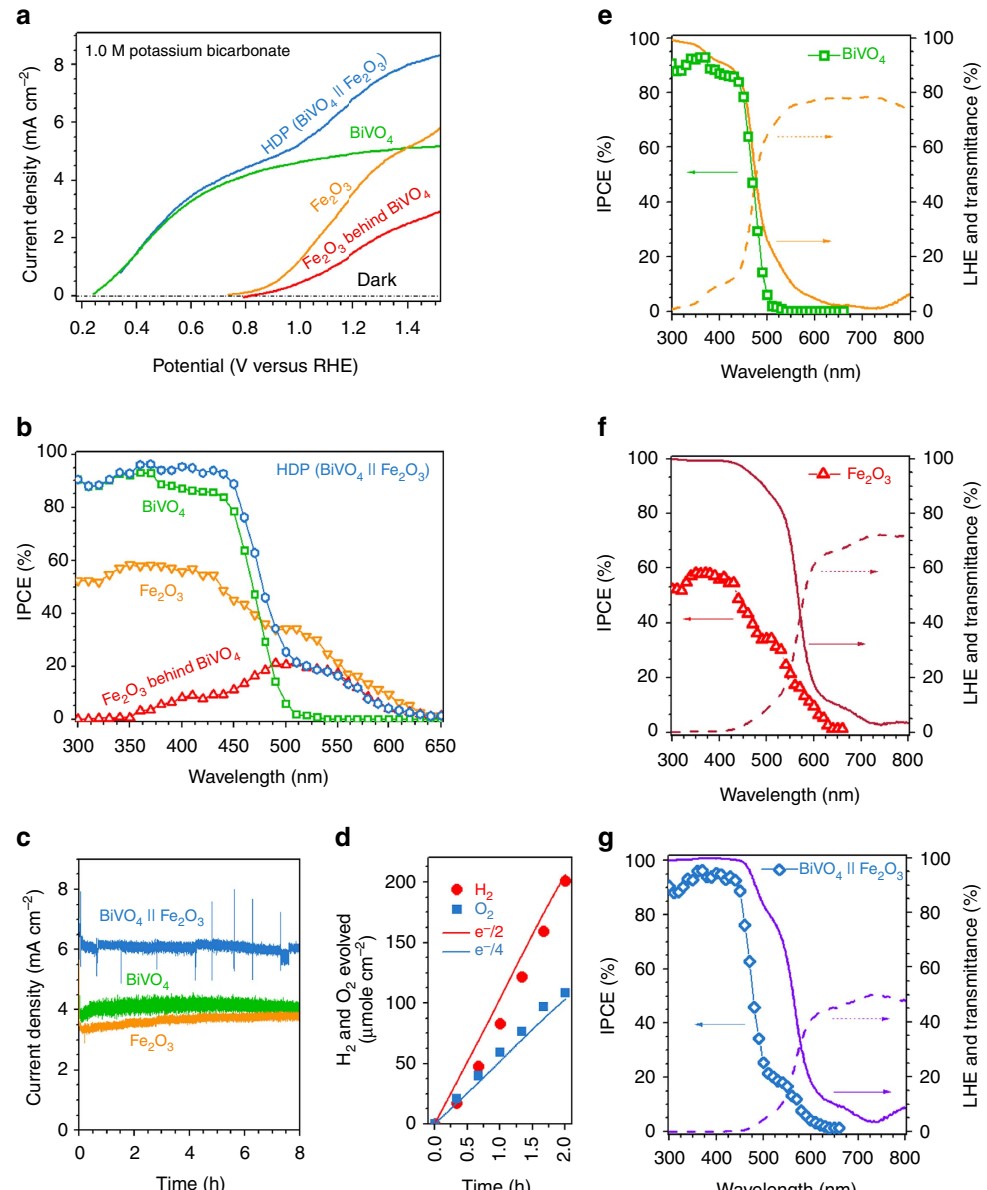

**Figure 3 | Solar water splitting in a biased PEC cell.** (**a**) Steady-state *I–V* curves. (**b**) IPCEs at 1.23 $V_{RHE}$. (**c**) Stability test and (**d**) gas evolution from PEC water splitting of photoanodes at 1.15 $V_{RHE}$. The solid lines in **d** are calculated values that correspond to the measured photocurrent. Action spectra for (**e**) $BiVO_4$, (**f**) $Fe_2O_3$ and (**g**) HDP ($BiVO_4\|Fe_2O_3$) showing correspondence between IPCE, light-harvesting efficiency (LHE) and transmittance. Photoanodes $BiVO_4$ and $Fe_2O_3$ denote NiOOH/FeOOH/H, 1% Mo:$BiVO_4$ and $Ni_2FeO_x$/H, $TiO_2$/0.5% Ti:$Fe_2O_3$, respectively. All data were obtained using an electrolyte of 1.0 M KCi at pH = 9.2 under 1 sun illumination.

HDP of $BiVO_4\|Fe_2O_3$, we obtained a water oxidation photo-current density of $7.0 \pm 0.2 \, mA \, cm^{-2}$ at 1.23 $V_{RHE}$, which sets a new performance benchmark for metal oxide photoanodes. As summarized in Fig. 5 and Supplementary Table 1, it is the first metal oxide photoanode to break the barrier of $7.0 \, mA \, cm^{-2}$ at 1.23 $V_{RHE}$. It is also meaningful that the result demonstrates how to move forward beyond $BiVO_4$ photoanode, which is the current forerunner material in solar water-splitting performance but is hampered by its relatively large bandgap. The HDP photoanode was successfully incorporated into a tandem cell with a c-Si solar cell for unbiased solar water splitting to demonstrate a stable and reproducible $\eta_{STH}$ of 7.7%. There are two easily conceivable ways to reach $\eta_{STH}$ higher than 10%, which is required for practical solar water splitting and the goal of most solar fuel research projects in progress worldwide. First, the saturated photocurrent of the current HDP is already over $8.3 \, mA \, cm^{-2}$ (Fig. 3a) and

thus $\eta_{STH} > 10\%$ ($8.1 \, mA \, cm^{-2}$) could be obtained by further reducing the onset potential of the photoanode. Second, if the bulk charge separation efficiency of the $Fe_2O_3$ electrode increases from its current 40% to a modest 50%, a photocurrent of $8.1 \, mA \, cm^{-2}$ or $\eta_{STH}$ of 10.1% is expected. Considering the rapid progress made in the last decade as depicted in Fig. 5, it seems highly likely that those two issues can be solved and that the goal of 10% $\eta_{STH}$ can be achieved within the foreseeable future. Hence, the HDP concept proposed here represents a significant step forward *en route* to practical solar hydrogen production.

## Methods

**Preparation of $BiVO_4$ films.** All chemicals used in this study were of analytical grade and used without further purification. $BiVO_4$ film was prepared by a modified metal-organic decomposition method with a slight modification from our previous procedure[8]. Thus, 0.2 M Bi(NO$_3$) · 5H$_2$O (99.8%; Kanto Chemicals)

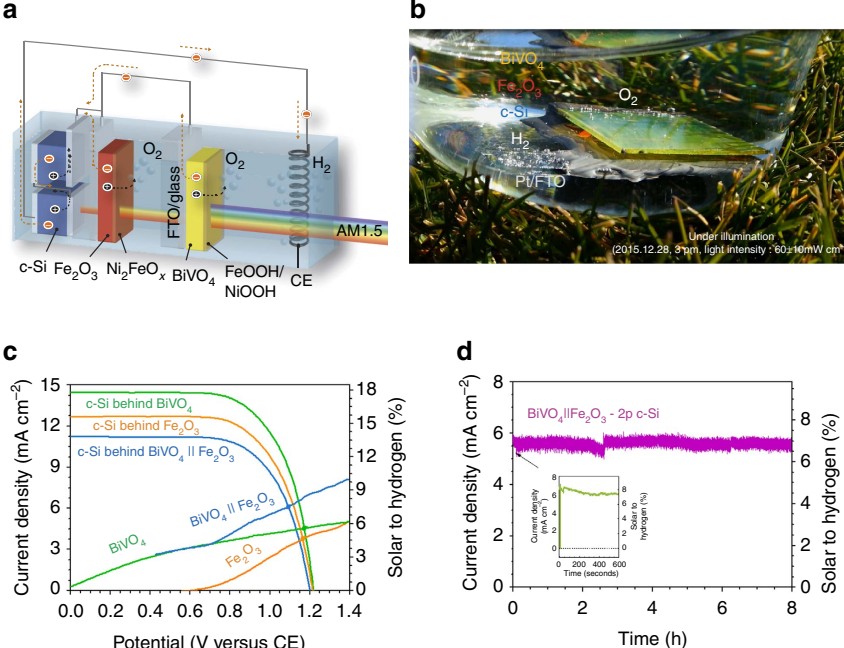

**Figure 4 | Unassisted solar water splitting with a HDP–PV (photovoltaic) device tandem cell.** (**a**) Scheme of a tandem cell with a HDP ($BiVO_4$||$Fe_2O_3$) and parallel-connected Si solar cells (2p c-Si), (**b**) artificial leaf (monolithic tandem cell) in action under illumination with real sunlight (details in Supplementary Movie 2) (**c**) Two-electrode AM 1.5 $I$–$V$ curves for photoanodes of $BiVO_4$, $Fe_2O_3$, $BiVO_4$||$Fe_2O_3$ and with a 2p c-Si solar cell behind each photoelectrode. (**d**) Stability of unbiased water splitting (0.0 V versus counter electrode). The small inset shows data recorded for short term (during stabilization of the tandem cell). Photoanodes were NiOOH/FeOOH/H, 1% Mo:$BiVO_4$ and $Ni_2FeO_x$/H, $TiO_2$/0.5% Ti:$Fe_2O_3$. All data were collected in an electrolyte of 1.0 M KCl, pH = 9.2.

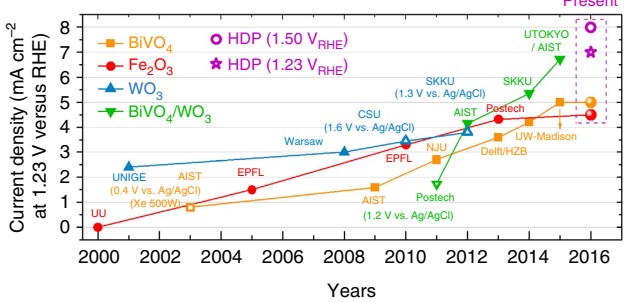

**Figure 5 | Reported photocurrents of metal oxide photoanodes.** HDP denotes $BiVO_4$||$Fe_2O_3$ hetero dual photoelectrodes, and the unfilled symbols indicate that photocurrent values were measured at a potential different from the standard 1.23 $V_{RHE}$. Numerical data are given in Supplementary Table 1.

dissolved in acetic acid (99.7%; Kanto Chemicals), 0.03 M VO(acac)$_2$ (98.0%; Sigma Aldrich) and 0.03 M MoO$_2$(acac)$_2$ (98.0%; Sigma Aldrich) in acetyl acetone (>99.0%; Kanto Chemicals) were prepared as a precursor solution. Then stoichiometric amount of each precursor was mixed to complete a precursor solution. For Mo doping, Bi:(V + Mo) = 1:1 atomic ratio was applied for 1% Mo:$BiVO_4$ films. For fabrication of a $BiVO_4$ film, 60 µl of solution was dropped on a fluorine doped tin oxide (FTO) glass (2 cm × 2.5 cm) and dried for 15 min in Ar atmosphere. Depending on synthesis condition, 10–80 µl of precursor was used. The FTO glass (TEC 8; Pilkington) was cleaned by using KOH (0.1 M) + ethanol with ratio of 1:5, and washed with copious amount of deionized water and finally stored in acetone before usage. The greenish transparent precursor film was calcined at 550 °C for 25 min to form a yellow $BiVO_4$ film. After annealing process, 2 cm × 2.5 cm $BiVO_4$/FTO was split to obtain photoanodes with a net irradiation area of 0.24 cm$^2$ connected by silver paste and copper wire and sealed with epoxy resin.

**Preparation of $Fe_2O_3$ films and $TiO_2$ surface treatment.** A haematite ($Fe_2O_3$) film was prepared by a polymer-assisted nitrate decomposition method. Thus, 0.5 M Fe(NO$_3$)$_2$·3H$_2$O (99.0%; Kanto Chemicals) was dissolved in

2-methoxyethanol (98.0%; Kanto Chemicals) and acetyl acetone (>99.0%; Kanto Chemicals). Volume ratio of 2-methoxyethanol and acetyl acetone was 7:3. Then, 500 mg of polyethylene glycol 8000 (polymer mass) (>99.0%; Sigma Aldrich) was added in 10 ml of precursor solution as a polymer binder. Prepared solution was sonicated for 1 h to achieve a homogeneous but opaque red wine-like coloured solution. Then, a Ti precursor solution was added as dopant with a Fe:Ti atomic ratio of 0.995 : 0.005 for 0.5% Ti concentration. The Ti precursor was 0.5 M Ti(IV) propoxide (99%; Sigma Aldrich) and 2-methoxyethanol (98.0%; Kanto Chemicals) solution. For the $Fe_2O_3$ film formation, 6 µl of the prepared solution were dropped on FTO glass (2 cm × 2.5 cm), and dried in Ar atmosphere for 15 min. After full drying (colour of precursor film became violet), the film was preheated at 80 °C for 10 min (the precursor film became brown) in Ar atmosphere and heat-treated at 500 °C for 10 min in the furnace. To obtain a desired film, the process was repeated for three times and finally the film was annealed at 500 °C for 75 min.

For $TiO_2$ surface treatment, 0.05 M titanium oxy acetylacetonate (90%; Sigma Aldrich) and 2-methoxyethanol (98.0%; Kanto Chemicals) solution was used. The as-prepared $Fe_2O_3$ film was coated with a Ti(IV) precursor by spin coating (1,000 r.p.m., 20 s, 2 times) and heat-treated at 500 °C for 10 min.

**Reduction treatment of metal oxide films.** Reduction treatment was conducted using a borohydride decomposition method. First, 16 mmol of NaBH$_4$ (>98%; Sigma Aldrich) was put in a 200 ml alumina crucible and another smaller alumina bottle (15 ml) was put on the NaBH$_4$ powder. In this smaller bottle, the as-prepared metal oxide film (2 cm × 2.5 cm) was placed and finally the 200 ml alumina crucible was covered with an alumina cover. This reactor was put in a preheated furnace at 500 °C for 30 min. Then the crucible was immediately taken out from the furnace and cooled down naturally.

**NiOOH/FeOOH co-catalyst deposition on the $BiVO_4$ film.** The NiOOH/FeOOH co-catalysts were deposited utilizing photo-assisted electrodeposition (PED) under AM 1.5G illumination. FeOOH was deposited from a 0.1 M Fe(SO$_4$)$_2$·7H$_2$O (≥99%; Sigma Aldrich) solution previously purged with Ar for 15 min at 0.25 V versus Ag/AgCl. Subsequent deposition was made with NiOOH from a 0.1 M Ni(SO$_4$)$_2$·6H$_2$O (99%; Sigma Aldrich) solution of pH ∼7.0 (by adding 0.1 M KOH) at 0.11 V versus Ag/AgCl. Optimum deposition times for FeOOH and NiOOH were 15 and 6 min, respectively. Before the PED process for FeOOH, the electrode was put on idle for 5 min, with vigorous stirring to minimize bubble attachment during the deposition process. For both co-catalysts, after PED process, constant current mode (0.03 mA cm$^{-2}$ recording a potential of ∼1.3 V versus Ag/AgCl and potential gradually went down) was used for 2 min in the dark to completely cover surface of electrodes (especially to cover possible pinholes).

**NiFeO$_x$ co-catalyst deposition on Fe$_2$O$_3$ film.** To deposit Ni$_2$FeO$_x$, Ni$_3$FeO$_x$ and NiFeO$_x$ co-catalysts on the Fe$_2$O$_3$ film, a single-solution method was used following Berlinguette et al.[29] for similar configuration. Precursor solutions of Ni(NO$_3$)$_2 \cdot$ 3H$_2$O (99.0%; Sigma Aldrich) and Fe(NO$_3$)$_2 \cdot$ 3H$_2$O (99.0%; Kanto Chemicals) were prepared with 2-methoxyethanol (98.0%; Kanto Chemicals) and mixtures of Ni:Fe of proper ratios were made. The mixture of Ni + Fe solutions was diluted to a concentration of 0.01 M before usage. For deposition of the electrocatalyst, 1 ml of the mixture solution was dropped on Fe$_2$O$_3$ photoelectrode (0.24 cm$^2$) and dried in air, and the photoelectrode was rinsed with 0.1 M KOH for 10 s.

For electrochemical analysis, above precursors with a thicker concentration (0.2 M) were used for NiO$_x$, FeO$_x$ and Ni$_2$FeO$_x$ preparation on FTO. The deposition procedure was the same as above except FTO electrode (0.5 cm$^2$) was used instead of photoelectrode.

**Loading additional co-catalysts on Fe$_2$O$_3$.** Cobalt phosphate (Co-Pi) was deposited by PED using illumination (100 mW cm$^{-2}$) at 0.4 V (versus Ag/AgCl) in 0.3 mM Co(NO$_3$)$_2 \cdot$ 6H$_2$O ($\geq$98%; Aldrich) and 100 ml of 0.5 M potassium phosphate (K$_2$HPO$_4$) at pH 7. A very small deposition photocurrent was observed during deposition ($\sim$15 µA cm$^{-2}$) and deposition was conducted for 120 s. FeOOH was deposited by PED at the same conditions used for BiVO$_4$ but with higher potential 0.45 V (versus Ag/AgCl) for 600 s.

**Fabrication of HDP (BiVO$_4$||Fe$_2$O$_3$)–c-Si tandem cell.** A tandem cell was made with HDP and c-Si solar cell, following a similar protocol used for our previous work with BiVO$_4$ photoanode and CH$_3$NH$_3$PbI$_3$ perovskite solar cell[8]. A piece of c-Si solar cell was cut in small pieces to make parallel connection (2p). Then, two pieces of c-Si were filed up on Al side (hole transfer side) and FTO (electron transfer side) with Ag paste. The area of two pieces became 0.30 cm$^2$ (0.5 cm × 0.6 cm). And BiVO$_4$ and Fe$_2$O$_3$ films were attached sequentially by glue with the c-Si solar cell at the bottom. Distance between BiVO$_4$ and Fe$_2$O$_3$ film was adjusted to 0.3 cm by glue and they were electrically attached by Ag paste and copper sheet to backside of 2p c-Si solar cell (Al side, hole transfer side). This made electron generated from HDP travels to hole transfer side of 2p c-Si solar cell. Finally, exposed FTO side of bottom 2p c-Si solar cell was connected to copper wire and all possible electrical connections were covered with epoxy (Locktit, cured in room temperature, 1 h) for isolation. For monolithic artificial leaf structure, much larger piece of c-Si solar cell and photoanode was prepared and final active area was 5.0 cm$^2$. As the counter electrode, Pt/FTO (1.0 cm$^2$) was prepared by spin coating of 0.5 mM H$_2$PtCl$_6 \cdot$ 6H$_2$O ($\geq$99%; Aldrich) diluted in ethanol (1,000 r.p.m., 10 s) on FTO and annealed for 30 min at 450 °C. This piece was directly connected on solar cell anode for electron intake.

**Characterization.** X-ray diffraction measurements were carried out with X-ray diffractometer using Ni-filtered Cu K$\alpha$ ($\lambda = 1.54178$ Å) radiation from a rotating anode source (X'Pert PRO MPD, PANalytical, 30 mA, 40 kV). Ultraviolet–visible absorbance was measured with a UV/Vis spectrometer (UV-2401PC, Shimadzu). As a reference, BaSO$_4$ powder attached on FTO was used. The morphology of the samples was observed using a field-emission (JEOL JMS-7400F, operated at 10 keV), and composition was examined by energy-dispersive X-ray spectroscopy (EDX). The chemical state of BiVO$_4$ and other films were probed by XPS with an ESCALAB 250Xi spectrometer. Detailed microscopic structure and corresponding energy-dispersive X-ray spectroscopy data were observed using Cs-corrected high-resolution scanning transmission electron microscope (JEOL, JEM 2200FS, 200 kV).

**Measurements of PEC performance.** PEC measurements of photoelectrodes were performed with a standard three-electrode configurations; photoanode as the working electrode, Pt mesh as the counter electrode and Ag/AgCl (3 M NaCl) as the reference electrode. The scan rate for the current–voltage ($J$–$V$) curve was 20 mV s$^{-1}$. For water oxidation experiments, 1.0 M potassium bicarbonate (KCi) electrolyte ($>$99.0%, Sigma Aldrich; pH $\sim$9.2) was used as the main electrolyte, and comparison experiment was conducted by using 1.0 M potassium phosphate (K$_2$HPO$_4$) buffer (pH $\sim$7.0) and 1.0 M KOH (pH 13.6). To measure the degree of charge separation, 1.0 M Na$_2$SO$_3$ ($>$98%, Sigma Aldrich) was added to the pH 7, 0.5 M phosphate buffer electrolyte. Potentials were recorded with correction by the Nernst relation $E_{RHE} = E_{SCE} + 0.0591$ pH + 0.209, in which $E_{Ag/AgCl}$ is applied bias potential and 0.209 is a conversion factor from the Ag/AgCl electrode to the reversible hydrogen electrode (revisible hydrogen electrode (RHE)) scale. All electrochemical data were recorded by using a potentiostat (IviumStat, Ivium Technologies). A 300 W Xenon lamp was used to make simulated 1 sun light irradiation condition (AM 1.5G, 100 mW cm$^{-2}$) by using a solar simulator (Oriel 91160) with an AM 1.5G filter calibrated with a reference cell certified by the National Renewable Energy Laboratories, USA.

IPCE measurement was conducted using the 300 W Xe lamp as the light source with liquid IR filter and a monochoromator (Oriel Cornerstone 130 1/8 m monochromator) with a bandwidth limit of 5 nm. The intensity of light was measured before IPCE measurements by photodiode detector (Oriel 70260).

Calculation of IPCE was carried out by the formula

$$IPCE(\%) = \frac{1,240 \times J}{\lambda \times P} \times 100 \qquad (1)$$

where $J$ = photocurrent density (mA cm$^{-2}$), $P$ = light power density (mW cm$^{-2}$) at $\lambda$, and $\lambda$ = wavelength of incident light (nm).

The MS plot was used to determine electrochemical properties using the equation;

$$\frac{1}{C^2} = \frac{2(V - V_f - kT/e)}{e\varepsilon\varepsilon_o N_D A^2} \qquad (2)$$

where $C$ = capacitance of photoanode (metal oxide + electrolyte double layer and so on), $e$ = charge of electron (C), $\varepsilon$ = dielectric constant of BiVO$_4$ and Fe$_2$O$_3$, $\varepsilon_o$ = permittivity of vacuum, $V$ = applied bias (versus RHE), $V_f$ = flat band potential (versus RHE), $k$ = Boltzmann constant, $N_D$ = donor density for $n$-type semiconductor (cm$^{-3}$), $A$ = surface area of photoanode and $T$ = temperature (K).

**PEC H$_2$ and O$_2$ evolution measurements.** Using Ar as a carrier gas, the amounts of H$_2$ and O$_2$ gases evolved from the PEC cell were analysed using a gas chromatograph (HP5890, molecular sieve 5l column) equipped with a thermal conductivity detector. Light source and electrolyte were the same as those used for above PEC measurements, and the gas products were sampled every 20 min.

**Characterization of BiVO$_4$ and α-Fe$_2$O$_3$ on doping and H$_2$ treatment.** The X-ray diffraction crystal structures of the photoanode films were standard monoclinic BiVO$_4$ and haematite α-Fe$_2$O$_3$ as shown in Supplementary Fig. 4. In the XPS spectra of Supplementary Fig. 5, the binding energies of Bi ($4f_{5/2}$ at 164.3 eV and $4f_{7/2}$ at 159.1 eV) and V ($2p_{1/2}$ at 523.5 eV and $2p_{3/2}$ at 515.9 eV) correspond to standard BiVO$_4$. A $\sim$0.3 eV shift to lower binding energies was observed after hydrogen treatment due to the partial reduction of Bi$^{3+}$ and V$^{5+}$, as reported previously for metal oxide semiconductors treated with H$_2$ or N$_2$ (refs 8,22). No clear shift is observed for Mo $3d$ ($3d_{3/2}$ at 235.9 eV and $3d_{5/2}$ at 232.6 eV) showed near absent change due to small XPS signal[8]. Fe $3d$ also showed a lower energy shift ($\sim$0.3 eV) as an indication of reduction. Interestingly, Ti $2p$, derived from dopant and surface treatment for Fe$_2$O$_3$ also showed a shift to lower binding energies ($\sim$0.4 eV). The binding energy of Ti $2p$ was lower than that of the standard TiO$_2$ ($2p_{3/2}$ at 459.0 eV)[30] because it was in a doped state or interaction with Fe$_2$O$_3$.

Effects of the combined doping and hydrogen treatment on the charge carrier density ($N_D$) were studied by MS analysis in Supplementary Fig. 8 in connection to PEC water oxidation performance. Thus, $N_D$ (cm$^{-3}$) was calculated assuming that the geometric area (1 cm$^{-2}$ in the present case) was the same as the actual surface area of material for all samples according to the equation;

$$N_D = \frac{2}{e\varepsilon\varepsilon_o} \left( \frac{d(1/C^2)}{dV} \right)^{-1} \qquad (3)$$

where $e$ = charge of electron (1.6 × 10$^{-19}$ C), $\varepsilon$ = dielectric constant of BiVO$_4$ ($\sim$68), $\varepsilon_o$ = permittivity of vacuum and $V$ = applied bias versus RHE[8,31,32]. In agreement with previous reports, Mo doping for BiVO$_4$, Ti doping for Fe$_2$O$_3$ and H$_2$ treatment for both metal oxides, decreased the slope of 1/$C^2$ indicating the increased charge carrier density[8,22,30,33,34]. Oxygen vacancies formed by hydrogen treatment introduce excess majority carriers as V$_o\cdot$, which acts as an $n$-type dopant in metal oxides[22,33–36]. The reduced XPS-binding energies for the metallic elements observed for BiVO$_4$ and Fe$_2$O$_3$ (Supplementary Fig. 5) are consistent with the reduced oxidation state one would expect for oxygen deficient metal oxides.

**Co-catalyst loading and the characterization.** To promote water oxidation further, oxygen-evolving co-catalysts were used. For BiVO$_4$, Co-Pi is known to be one of the most active co-catalysts[8,19], but its stability is poor. Hence, NiOOH/FeOOH was selected here, and our NiOOH/FeOOH/H, 1% Mo:BiVO$_4$ photoanode showed a photocurrent of 5.0 ± 0.2 mA cm$^{-2}$ at 1.23 V$_{RHE}$. This is close to the state of the art, and corresponds to an applied bias photon-to-current efficiency (ABPE) of 2.08% as shown in Supplementary Fig. 9, which is comparable to the highest value recorded for metal oxide photoelectrodes (NiOOH/FeOOH/N$_2$:BiVO$_4$, ABPE of 2.2% (ref. 22)). For Fe$_2$O$_3$, Ni$_2$FeO$_x$ showed the highest activity among co-catalysts tested in this work, including Co-Pi, FeOOH, NiFeO$_x$ and Ni$_3$FeO$_x$. In particular, Ni$_2$FeO$_x$ produced by our new synthesis method using drop casting of a nitrate salt solution and in situ activation during PEC operation showed the best results. XPS analysis in Supplementary Fig. 5 indicates that the NiOOH/FeOOH co-catalyst has oxidation states close to Fe$^{3+}$ and Ni$^{2+/3+}$ as reported[37]. The oxidation state of Ni in Ni$_2$FeO$_x$ is also likely to be mixed Ni$^{2+/3+}$ (refs 10,38). The O1s signal for the NiOOH/FeOOH-modified photoanodes shows a pronounced extra peak at 531.3 eV, which confirms that the NiOOH/FeOOH and Ni$_2$FeO$_x$ co-catalysts are in the oxy hydroxide form[10,39].

**Surface modification of α-Fe$_2$O$_3$ and the PEC water-splitting performance.** Recombination at surface states is a well-known issue for unmodified Fe$_2$O$_3$ photoanodes (Supplementary Fig. 13). In addition to lowering the photocurrent, these states can also cause Fermi level pinning, which causes part of the potential

drop to fall across the Helmholtz layer. This leads to a lower photovoltage ($V_{Ph}$) and an undesired positive shift of the photocurrent onset potential[40]. Recently, there have been many successful examples of passivating the surface states of haematite by depositing metal oxide overlayers[25,40–42]. Here we passivate these surface states by depositing a thin $TiO_2$ layer on the surface of hydrogen-treated 0.5% $Ti:Fe_2O_3$ using a facile solution-based method. As shown in Supplementary Fig. 13, this increases the photocurrent density markedly from 0.88 to 3.0 mA at 1.23 $V_{RHE}$ in a 1.0 M NaOH electrolyte and shifted the onset potential cathodically by ~115 mV. There was no obvious change in the carrier density of haematite on $TiO_2$ loading and the flat band potentials are similar, as demonstrated by the MS plots (Supplementary Fig. 8e,f). Thus, our optimized $Ni_2FeO_x$-catalysed, hydrogen-treated and $TiO_2$-modified 0.5% $Ti:Fe_2O_3$ photoanodes showed photocurrents of 3.5–4.1 mA cm$^{-2}$ at 1.23 $V_{RHE}$ in a pH 9.2 KCi electrolyte, which is the amongst the highest ever recorded for $Fe_2O_3$-based photoanodes in near-neutral electrolytes, and comparable with reported top-performing $NiFeO_x/Fe_2O_3$ (ref. 4), $NiFeO_x/Al_2O_3/Si:Fe_2O_3$ (ref. 10) photoanodes. Interestingly, loading of the $Ni_2FeO_x$ co-catalyst on $Fe_2O_3$ photoanode significantly reduced the performance gap between pH 9.2 (1.0 M KCi) and pH 13.6 (1.0 M KOH) (Supplementary Fig. 14). In the meantime, it would be more effective for practical application as water oxidation efficiency (especially $Fe_2O_3$) can be maximized if HDP can be operated in highly basic solution, owing to pH gradient problem in mild electrolyte condition[43].

**Data availability.** The data that support the findings of this study are available from a corresponding author (J.S.L.) on request.

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

## Acknowledgements

This research was supported by the Climate Change Response project (2015M1A2A2074663 and 2015M1A2A2056824), the Basic Science Grant (NRF-2015R1A2A1A10054346), Korea Center for Artificial Photosynthesis (KCAP, No. 2009-0093880) funded by MSIP, Project No. 10050509 funded by MOTIE of Republic of Korea and the PECDEMO project (co-funded by Europe's Fuel Cell and Hydrogen Joint Undertaking under Grant Agreement no. 621252).

## Author contributions

J.H.K. and J.-W.J. conceived the HDP concept; R.v.d.K. and J.S.L defined and supervised the project; J.H.K prepared the $BiVO_4$, $Fe_2O_3$ photoanodes and the photoanode–Si tandem cell, and conducted characterization of materials; Y.H.J. prepared and characterized the series-arranged Si solar cell; Y.H.L arranged the video of the artificial leaf acting under simulated and actual sun illumination; F.F.A helped to analyse the data; J.H.K, J.-W.J., R.v.d.K. and J.S.L co-wrote the manuscript. All authors viewed and commented on the manuscripts.

## Additional information

**Competing financial interests:** The authors declare no competing financial interests.

