## [Peer Review File · Nature Communications]

Reviewers' comments:

Reviewer #1 (Remarks to the Author):

This paper looks at a dual photoanode PEC device that utilizes BiVO₄ and Fe₂O₃ as the anodes. They couple their dual anodes with 2 Si solar cells to create an unbiased water splitting device that operates at 7.7% STH efficiency. They use BiVO₄ as the top absorber, then hematite, then the Si solar cells. The hematite and BiVO₄ are in a pH 9.2 potassium bicarbonate buffer that allows them to operate the two photoanodes in the same solution. Each of the photoanodes uses a NiFe OER catalyst, with BiVO₄ using the LDH oxy hydroxide catalyst as reported by Choi on BiVO₄ photoanodes, and hematite using an amorphous NiFe catalyst as reported by Berlinguetta and others. In addition, they use a 1% Mo doping for BiVO₄ and a 0.5% Ti doping for hematite, a TiO₂ passivation layer on hematite, and they anneal each photoanode in H₂ to increase the carrier concentration, all previously reported phenomena.

The novelty of this paper is using the dual photoanode system. It is a clever use of cell design that results in the highest STH efficiency for earth quasi stable materials. While no new understanding is generated around the doping, catalyst-photoanode interface, or material preparation, they utilize the state of the art for hematite and BiVO₄ to demonstrate the progress of the field as a whole, which I feel is significant. Nevertheless, it is important to point out that I would not call BiVO₄ "earth abundant", given the scarcity of Bi (on the same level as Ag and Se).

The study is carried out carefully and is fairly well written. My biggest complaint would come from their misleading and gimmicky title. Upon reading the title I expected that their study would generate a greater understanding or correlation between seaweed and artificial photosynthesis, but really it doesn't have any place in this study other than a cheap way to generate excitement. I think it would be safe to take out of their manuscript entirely without harming the content.

I think they can also remove or edit figure 2D from their main text. It is confusing to understand and there is no explanation for this figure in the main text. One has to dig through the SI in order to understand what the relationship between the efficiency and the microliters is.

Additional technical comments:

1. There are two typos in SI: plot d and f in XPS (Figure S5) should be assigned to Fe 2p not Fe 3d.
2. I am also curious about the feature of the JV curve of BiVO₄||Fe₂O₃ in Figure 3a. Was the abrupt increase of photocurrent at 1.0 V vs. RHE attributed to integration of Fe₂O₃?
3. In terms of JV curve of 0.5% Ti: Fe₂O₃ in Figure S6 and S7, could the authors explain more about the anodic peak observed at around 1.0 V?
4. One of the highlights of this manuscript is the high η_{STH} of 7.7 % in unassisted water splitting system, however, the efficiency here is also highly depending on the performance of photocathode. The authors adopted a Si solar cell which could offer up to 1.2 V photovoltage, which weakened the application of hetero-type dual photoanode.

Reviewer #2 (Remarks to the Author):

The manuscript by Jin Hyun Kim and Ji-Wook Jang et al. titled "Natural seaweeds-inspired hetero-type dual photoanodes for unbiased solar water splitting with extended light harvesting" demonstrates a relatively simple strategy to boost photocurrent and increase solar-to-hydrogen water splitting efficiencies by employing tandem photoanodes, optimized for different parts of the solar spectrum, and connected in parallel in order to increase the spectral range of light harvesting without the need for current matching, as required in conventional heterojunction photoelectrodes,

although modest tradeoffs with respect to the optimal operating pH for each photoanode are required. Inspired by the optimized spectral response of seaweeds that evolved at different ocean depths, this general strategy to boost efficiency has been successfully employed in an analogous manner in several other photovoltaic and photonic up-conversion systems unrelated to water splitting, and the results presented in this manuscript clearly demonstrate that the strategy can also be greatly beneficial to efficiencies in unbiased solar water splitting. The body of the manuscript is well-written with convincing data and figures, and given the relatively poor charge separation efficiency of current hematite electrodes, future advances in terms of more efficient Fe₂O₃ electrodes should lead to even greater enhancements in performance with the tandem dual photoanode architecture employed by the authors. I recommend publication of the manuscript after the following minor issues associated with the materials and methods section and supporting information are addressed by the authors:

- 1) At numerous points in the materials and methods section, the authors refer to the use of "2-methyl methanol" as a solvent used in the preparation of the electrodes. As this is not a valid chemical compound, the authors should clarify what solvent they are actually using, preferably with IUPAC notation. Also, on the top of page 25, the authors refer to nonstoichiometric "KHPO₄" instead of the intended K₂HPO₄.
- 2) In section S1 of the supplementary text, the authors refer to the "Fe 3d" peaks in their XPS data instead of the intended "Fe 2p" peaks shown in their data. Similarly, the label in the upper left-hand corner of Figure S5a incorrectly labels the spectra as "Bi (4s)" instead of "Bi (4f)". In the figure caption, sections d and f are also incorrectly labeled as "Fe 3d" instead of "Fe 3p".
- 3) Although the thin dotted lines in Figures S1 and S13 are presumably dark scans, they are not labeled as such in the figure or in the caption.
- 4) In Figure S9, "Transmittance" is misspelled as "Tranmittance" in panels a and b, and there are several errors - formatting and otherwise - in the supplementary references. For example, see references S31, S47, S53, and S54.
- 5) The optimization discussion after Figure S11 in the supplementary text requires dramatic improvements in grammar and sentence structure prior to publication.

Reviewer #1

This paper looks at a dual photoanode PEC device that utilizes BiVO₄ and Fe₂O₃ as the anodes. They couple their dual anodes with 2 Si solar cells to create an unbiased water splitting device that operates at 7.7% STH efficiency. They use BiVO₄ as the top absorber, then hematite, then the Si solar cells. The hematite and BiVO₄ are in a pH 9.2 potassium bicarbonate buffer that allows them to operate the two photoanodes in the same solution. Each of the photoanodes uses a NiFe OER catalyst, with BiVO₄ using the LDH oxy hydroxide catalyst as reported by Choi on BiVO₄ photoanodes, and hematite using an amorphous NiFe catalyst as reported by Berlinguetta and others. In addition, they use a 1% Mo doping for BiVO₄ and a 0.5% Ti doping for hematite, a TiO₂ passivation layer on hematite, and they anneal each photoanode in H₂ to increase the carrier concentration, all previously reported phenomena.

The novelty of this paper is using the dual photoanode system. It is a clever use of cell design that results in the highest STH efficiency for earth quasi stable materials. While no new understanding is generated around the doping, catalyst-photoanode interface, or material preparation, they utilize the state of the art for hematite and BiVO₄ to demonstrate the progress of the field as a whole, which I feel is significant.

Comments 1: *Nevertheless, it is important to point out that I would not call BiVO₄ "earth abundant", given the scarcity of Bi (on the same level as Ag and Se).*

Response 1: We totally agree that Bi is not an earth abundant element. Here in this manuscript we did not specify BiVO₄ but tried to make a general statement that **many** metal oxides are composed of earth abundant elements.

*"In addition, **many** metal oxides are composed of earth-abundant elements" (on Page 3)*

Comments 2: *The study is carried out carefully and is fairly well written. My biggest complaint would come from their misleading and gimmicky title. Upon reading the title I expected that their study would generate a greater understanding or correlation between seaweed and artificial photosynthesis, but really it doesn't have any place in this study other than a cheap way to generate excitement. I think it would be safe to take out of their manuscript entirely without harming the content.*

Response 2: Indeed, we agree with the referee's point of view on the limited relevance of seaweeds and the core of our work. In order to accommodate the reviewer's point while noting the interesting similarity between the two systems in the sense of wavelength-selective solar light absorption, we have revised the whole manuscript as follows:

- i) We have removed 'natural seaweeds-inspired' from the title as well as the whole text of abstract, results, and discussion throughout the manuscript.

- ii) We have modified the manuscript such that the concept of hetero-type dual photoanode (HDP) is emphasized and have introduced the similarity with sea weeds in the sense of wavelength-selective solar light absorption for better understanding of the HDP concept.

Title: *“Natural seaweeds-inspired Hetero-type dual photoanodes for unbiased solar water splitting with extended light harvesting”*

Revised (on Page 4-5)

~~*“In search for an efficient light harvesting method for solar water splitting with two semiconductors of different band gaps, we note that natural seaweeds (or marine algae) develop varying colors depending on the depth of the sea that they inhabit. Photons with short wavelengths (<420 nm, blue) can penetrate deep into the sea water where long wavelength photons of low energy (>600 nm, red) cannot reach. Adapting to availability of light of different wavelengths, the seaweeds develop selective light harvesting capability for photosynthesis as indicated by evolution of their colors from green (Chlorophyta), yellow (Phaeophyceae), and then to red (Rhodophyta) as one goes deeper into the sea¹⁵. Instead of developing an ‘ideal’ single light absorber that can handle all range of photons, the seaweeds develop smart ‘wavelength optimized’ light absorbers for sequential light absorption. This simple example in nature inspired us to introduce a subtle but effective concept of a hetero-type dual photoelectrode (HDP) schematically presented in comparison with the natural seaweeds in Fig. 1.*~~

“In order to enhance the light harvesting in two semiconductors of different band gaps, here we introduce subtle but effective concept of a hetero-type dual photoelectrode (HDP) schematically presented in Fig. 1a. The HDP device contains two independent liquid-semiconductor junctions, in contrast to a conventional heterojunction photoelectrode that combines a single liquid-semiconductor junction and a solid-solid junction into a single photoelectrode. The advantage of our HDP concept is that we can independently optimize each photoelectrode and the performance of the HDP device becomes the simple sum of the individual performance of the two. If the large band gap semiconductor is used as the top absorber and the small band gap semiconductor at the bottom, both thermalization and non-absorption losses could be minimized. Another important advantage is, not like an n-type/p-type tandem photoelectrode system, that current-matching of the two electrodes is not required, which is a critical – and sometimes difficult to fulfil – requirement for multi-junction solar cells⁹ and heterojunction photoelectrodes. As a result, the spectral range of light harvesting in the solar spectrum utilized for PEC water splitting is extended, which can lead to enhanced photocurrents.

We have noted that the HDP concept is analogous to a natural photosynthesis system in the sense of wavelength-selective solar light absorption. Thus natural seaweeds (or marine algae) develop varying colors depending on the depth of the sea that they inhabit. It is owing to availability of different photons depending on depth of sea – red light can reach only shallow depth of sea because of low energy of photons (wavelength of >600 nm) while blue light (<420 nm) can reach deeper. Adapting to availability of photons with varying wavelengths, the seaweed colony is capable of selective light utilization for photosynthesis by varying their

habitat depth from green (Chlorophyta), yellow (Phaeophyceae), and then to red (Rhodophyta) as one goes deeper into the sea²¹. Instead of developing an ‘ideal’ single light absorber that can handle all range of photons, the seaweeds develop smart ‘wavelength optimized’ light absorbers for sequential light utilization. In Fig. 1b, the seaweed system is schematically presented to show the similarity with the HDP configuration. Such resemblance is rather interesting, as the other commonly used light utilization scheme – so called tandem cell also resembles PSII/PSI of natural photosynthesis in using multiple light absorbers.”

And we have exchanged the position of Fig. 1a and Fig. 1b to stress the importance of the HDP concept.

Revised (Figure 1)

Fig. 1. Wavelength-selective solar light absorption by hetero-type dual photoanode (HDP) vs. natural seaweeds. (a) HDP made with different band gap materials (e.g., BiVO_4 and Fe_2O_3). (b) Distribution of seaweeds, chlorophyta (green algae), phaeophyceae (brown algae) and rhodophyta (red algae) with depth of sea that absorb different parts of solar spectrum.

Comments 3: I think they can also remove or edit figure 2D from their main text. It is confusing to understand and there is no explanation for this figure in the main text. One has to dig through the SI in order to understand what the relationship between the efficiency and the microliters is.

Response 3: We thank reviewer for finding missing parts in the manuscript. We have fixed Fig. 2D, which now has a label of ‘Amount of BiVO_4 deposition ($\mu\text{l}/5.0 \text{ cm}^2$)’ and have added simple

explanation showing the relationship between the current density and the microliters (μl) in the revised manuscript and the caption of Fig. 2, as follows.

Revised (on Page 8)

“These photocurrents are optimized values by adjusting the thickness of BiVO_4 and Fe_2O_3 , which can be easily regulated by changing the quantity of precursor solution (μl). (See Fig. 2d, Supplementary Figs. 10, 11 and ‘Materials and Methods’ part for details.)”

Revised (Figure 2)

Fig. 2. BiVO_4 and Fe_2O_3 hetero-type dual photoelectrode (HDP, $\text{BiVO}_4 \parallel \text{Fe}_2\text{O}_3$) in a sacrificial sulfite solution. (Aa) Schematic working principle of a photoelectrochemical cell with HDP $\text{BiVO}_4 \parallel \text{Fe}_2\text{O}_3$ as the photoanode. (Bb) Photographs of fabricated electrodes showing a good transparency. (Cc) Steady-state I-V behaviors. (Dd) Optimization of the loading amount of BiVO_4 and Fe_2O_3 precursor solution (μl).

on 5.0 cm^2 of FTO glass loadings for the best photocurrent generation (See also Supplementary Fig. S11). (Ee) Incident photon-to-current efficiency (IPCE). (Ef) Utilization of light in AM 1.5 G spectrum by different photoanodes. Analyses were conducted in 0.5 M KPi and 0.5 M Na_2SO_3 of pH = 7.0.

Comments 4: There are two typos in SI: plot d and f in XPS (Figure S5) should be assigned to Fe 2p not Fe 3d.

Response 4: We thank the reviewer for finding our mistakes. We have corrected two typos in the Supplementary Figure S5 accordingly.

Revised (Supplementary Figure 5)

Figure S5. X-ray photoelectron spectra (XPS) of (a) Bi 4f, (b) V 2p and (c) Mo 3d of 1% Mo:BiVO₄ and H, 1% Mo: BiVO₄. (d) Fe ~~3d~~ 2p, (e) Ti 2p of 0.5% Ti: Fe₂O₃, H, 0.5% Ti: Fe₂O₃, TiO₂/0.5% Ti: Fe₂O₃ and H, TiO₂/0.5% Ti: Fe₂O₃. (f) Fe ~~3d~~ 2p, (g) Ni 2p, (h, i) O 1s of Ni₂FeO_x/H, TiO₂/0.5% Ti: Fe₂O₃ and NiOOH/FeOOH/H, 1% Mo:BiVO₄ (All peak intensities have been normalized).

Comments 5: I am also curious about the feature of the JV curve of BiVO₄//Fe₂O₃ in Figure 3a. Was the abrupt increase of photocurrent at 1.0 V vs. RHE attributed to integration of Fe₂O₃?

Response 5: Yes, the reviewer is right. In Fig. 3a, the photocurrent of HDP is the sum of those of BiVO₄ and Fe₂O₃ behind BiVO₄. It needs to be noted that the IV shape of HDP is similar to that of BiVO₄ at the potentials below 1.0 V_{RHE} and it follows well that of Fe₂O₃ behind BiVO₄ at the potentials above 1.0 V_{RHE}. Thus, the abrupt increase of photocurrent around at 1.0 V_{RHE} is due to the contribution of added photocurrent from Fe₂O₃ photoanode behind BiVO₄.

Comments 6: In terms of JV curve of 0.5% Ti: Fe₂O₃ in Figure S6 and S7, could the authors explain more about the anodic peak observed at around 1.0 V?

Response 6: JV curve of 0.5% Ti: Fe₂O₃ in Figure S6 and S7 reached its plateau around at 1.0V_{RHE}, which means that charge separation by band bending reached its limit (thus, no significant increment is observed by applying additional bias). We like to say that the shape of the curve is not really sharpening out to be called “a peak”. Please see Figure S1 for magnified image of IV curve for 0.5% Ti:Fe₂O₃ and Figure S6 for bulk charge separation efficiency of the electrode, which also follows the shape of IV curves. Additionally, in the water oxidation

reaction by the same electrodes (0.5% Ti: Fe₂O₃, and surface modified 0.5% Ti: Fe₂O₃), it is hard to find the anodic peak (Figure S13).

Comments 7: *One of the highlights of this manuscript is the high η_{STH} of 7.7 % in unassisted water splitting system, however, the efficiency here is also highly depending on the performance of photocathode. The authors adopted a Si solar cell which could offer up to 1.2 V photovoltage, which weakened the application of hetero-type dual photoanode.*

Response 7: We thank the reviewer for bringing up this important point. It is true that high applied voltage is necessary to achieve higher performance, which could limit the performance of the current HDP system. The positive onset potential of Fe₂O₃ is now critical bottleneck to increase the performance. However, we observed much improved onset potential of Fe₂O₃ by the simple surface modification (Fig. S13). We believe that further intensive optimization could result in early saturated photocurrent of HDP, which makes the system more applicable at lower voltages. Alternatively, use of a basic electrolyte (pH=13.6) can be a good option to improve the onset potential of Fe₂O₃ (Fig. S13) upon stabilizing BiVO₄ photoanodes in the same solution. Finally, the photovoltage higher than 1.0 V could be obtained with cheap solar cells like perovskite solar cells or dye sensitized solar cells.

To show that the performance of HDP system could be improved further, following statements were made in Discussion part of the main manuscript.

on Page 11, Discussion part

“The HDP photoanode was successfully incorporated into a tandem cell with a c-Si solar cell for unbiased solar water splitting to demonstrate a stable and reproducible η_{STH} of 7.7 %. There are two easily conceivable ways to reach η_{STH} higher than 10 %, which is required for practical solar water splitting and the goal of most solar fuel research projects in progress worldwide. First, the saturated photocurrent of the current HDP is already over 8.3 mA/cm² (Fig. 3A) and thus $\eta_{STH} > 10$ % (8.1 mA/cm²) could be obtained with reduced onset potential of the photoanode. Second, if the bulk charge separation efficiency of the Fe₂O₃ electrode increases from current 40% to modest 50%, a photocurrent of 8.1 mA/cm² or η_{STH} of 10.1 % is expected. Considering the rapid progress made in the last decade as depicted in Fig. 5, those two issues could be solved and the goal of 10 % η_{STH} is achieved in a foreseeable future. Hence, the HDP concept proposed here represents a significant step forward en route to practical solar hydrogen production.”

Figure S13. Effects of surface passivation by overlayer and electrolyte pH for Fe₂O₃ based photoanodes. (a) Schematics showing the effect of surface states and passivation by an overlayer. (b) 1.0 M KPi, pH 7.0, (c) 1.0 M KCl, pH 9.2 and (d) 1.0 M KOH, pH 13.6. (e) I-V curves of Ni_2FeO_x catalyzed and uncatalyzed $H, TiO_2/0.5\% Ti:Fe_2O_3$. Vertical line of red (1.0 M KOH), blue (1.0 M KCl) and black (1.0 M KPi) presents position of 1.23 V_{RHE} with conversion of NHE potential scale.

Reviewer #2

The manuscript by Jin Hyun Kim and Ji-Wook Jang et al. titled "Natural seaweeds-inspired hetero-type dual photoanodes for unbiased solar water splitting with extended light harvesting" demonstrates a relatively simple strategy to boost photocurrent and increase solar-to-hydrogen water splitting efficiencies by employing tandem photoanodes, optimized for different parts of the solar spectrum, and connected in parallel in order to increase the spectral range of light harvesting without the need for current matching, as required in conventional heterojunction photoelectrodes, although modest tradeoffs with respect to the optimal operating pH for each photoanode are required. Inspired by the optimized spectral response of seaweeds that evolved at different ocean depths, this general strategy to boost efficiency has been successfully employed in an analogous manner in several other photovoltaic and photonic up-conversion systems unrelated to water splitting, and the results presented in this manuscript clearly demonstrate that the strategy can also be greatly beneficial to efficiencies in unbiased solar water splitting. The body of the manuscript is well-written with convincing data and figures, and given the relatively poor charge separation efficiency of current hematite electrodes, future advances in terms of more efficient Fe_2O_3 electrodes should lead to even greater enhancements in performance with the tandem dual photoanode architecture employed by the authors. I recommend publication of the manuscript after the following minor issues associated with the materials and methods section and supporting information are addressed by the authors:

Comments 1: *At numerous points in the materials and methods section, the authors refer to the use of "2-methyl methanol" as a solvent used in the preparation of the electrodes. As this is not a valid chemical compound, the authors should clarify what solvent they are actually using, preferably with IUPAC notation. Also, on the top of page 25, the authors refer to nonstoichiometric " KHPO_4 " instead of the intended K_2HPO_4 .*

Response 1: We thank the reviewer for finding these important errors. We have revised them accordingly.

- 1) *Previous* :2-methyl methanol
Corrected: 2-methoxyethanol (which is also called as ethylene glycol monomethyl ether)
- 2) *Previous* : KHPO_4
Corrected: K_2HPO_4

Comments 2: *In section S1 of the supplementary text, the authors refer to the "Fe 3d" peaks in their XPS data instead of the intended "Fe 2p" peaks shown in their data. Similarly, the label in the upper left-hand corner of Figure S5a incorrectly labels the spectra as "Bi (4s)" instead of "Bi (4f)". In the figure caption, sections d and f are also incorrectly labeled as "Fe 3d" instead of "Fe 3p".*

Response 2: We thank the reviewer for finding these typos. We have revised Bi 4s to Bi 4f in the Supplementary Fig 6a and modified Fe 3d to Fe 2p in the figure captions of Supplementary Fig 6d, f.

Figure S5. X-ray photoelectron spectra (XPS) of (a) Bi 4f, (b) V 2p and (c) Mo 3d of 1% Mo:BiVO₄ and H, 1% Mo: BiVO₄. (d) Fe 3d 2p, (e) Ti 2p of 0.5% Ti: Fe₂O₃, H, 0.5% Ti: Fe₂O₃, TiO₂/0.5% Ti: Fe₂O₃ and H, TiO₂/0.5% Ti: Fe₂O₃. (f) Fe 3d 2p, (g) Ni 2p, (h, i) O 1s of Ni₂FeO₄/H, TiO₂/0.5% Ti: Fe₂O₃ and NiOOH/FeOOH/H, 1% Mo:BiVO₄ (All peak intensities have been normalized).

Comments 3: Although the thin dotted lines in Figures S1 and S13 are presumably dark scans, they are not labeled as such in the figure or in the caption.

Response 3: Thank you for reviewer's comment. We have added denotation that dotted lines represent dark currents throughout Supplementary Figures (Supplementary Figure 1, 8, 10, 12, 13)

Comments 4: In Figure S9, "Transmittance" is misspelled as "Tranmittance" in panels a and b, and there are several errors - formatting and otherwise - in the supplementary references. For example, see references S31, S47, S53, and S54.

Response 4: We are sorry for the frequent errors. We have changed "Transmittance" to "Transmittance" in the Supplementary Fig 10 as follows. And we have corrected all the errors in the supplementary references.

Figure S10. (a) Light harvesting efficiency (LHE), transmittance (T) of BiVO₄ and Fe₂O₃, and light absorbance of Fe₂O₃ from transmitted light through BiVO₄. (b) LHE corresponding to AM 1.5G spectrum.

We also have revised the supplementary reference list properly.

Previous

31. Sayama, K. *et al.* Effect of Carbonate Ions on the Photooxidation of Water over Porous BiVO₄ Film Photoelectrode under Visible Light. *Chem. Lett.* **39**, 17-19 (2010).
47. Miller, N. G. Y. C. J. K. A. D. E. L. Status of research on tungsten oxide-based photoelectrochemical devices at the University of Hawai'i. *SPIE 7770* (2010).
53. Luo, J. *et al.* Targeting Ideal Dual-Absorber Tandem Water Splitting Using Perovskite Photovoltaics and CuInxGa1-xSe2 Photocathodes. *Adv. Energy Mater.* **5**, n/a-n/a, doi:10.1002/aenm.201501520 (2015).
54. Dias, P. *et al.* Transparent Cuprous Oxide Photocathode Enabling a Stacked Tandem Cell for Unbiased Water Splitting. *Adv. Energy Mater.* **5**, n/a-n/a, doi:10.1002/aenm.201501537 (2015).

Corrected

2. Sayama, K. *et al.* Effect of Carbonate Ions on the Photooxidation of Water over Porous BiVO₄ Film Photoelectrode under Visible Light. *Chem. Lett.* **39**, 17-19 (2010).
22. Miller, N. G. Y. C. J. K. A. D. E. L. Status of research on tungsten oxide-based photoelectrochemical devices at the University of Hawai'i. *SPIE* **7770** (2010).
29. Luo, J. *et al.* Targeting Ideal Dual-Absorber Tandem Water Splitting Using Perovskite Photovoltaics and CuIn_xGa_{1-x}Se₂ Photocathodes. *Adv. Energy Mater.* **5**, 1-8, doi:10.1002/aenm.201501520 (2015).
30. Dias, P. *et al.* Transparent Cuprous Oxide Photocathode Enabling a Stacked Tandem Cell for Unbiased Water Splitting. *Adv. Energy Mater.* **5**, 1-8, doi:10.1002/aenm.201501537 (2015).

Comments 5: *The optimization discussion after Figure S11 in the supplementary text requires dramatic improvements in grammar and sentence structure prior to publication.*

Response 5: We are sorry for the unpolished sentences. We have corrected the manuscript for grammar and sentence structure, and removed unnecessary phrases in the note.

Revised (Notes below Supplementary Figure S11)

“Optimization of BiVO₄ and Fe₂O₃ used in this study was conducted for both individual photoanodes and HDP. Obviously, a lower amount of deposition granted a larger transmittance but a smaller LHE, whereas a higher amount of deposition gave results with opposite trend. Increasing amounts of BiVO₄ deposition granted progressively higher performance for HDP till transmittance of BiVO₄ was compromised at a deposition amount of 80 μl per 5.0 cm². (Fig. 1d). While the highest photocurrent for BiVO₄ was obtained for 80 μl per 5.0 cm² (5.40 mA/cm² at 1.23 V_{RHE}), significantly deterred transmittance resulted in lower photons available for underlying Fe₂O₃ (Fig. S11).”

REVIEWERS' COMMENTS:

Reviewer #1 (Remarks to the Author):

The authors have made satisfactory revisions. The manuscript should be published without further delays.